# DISENTANGLED FEATURE SWAPPING AUGMENTATION FOR WEAKLY SUPERVISED SEMANTIC SEGMENTATION

## ABSTRACT

Weakly supervised semantic segmentation utilizes a localization map obtained from a classifier to generate a pseudo-mask. However, classifiers utilize background cues to predict class labels because of a biased dataset consisting of images, in which specific objects frequently co-occur with certain backgrounds. Consequently, the classifier confuses the background with the target objects, resulting in inaccurate localization maps. To this end, we proposed a disentangled feature swapping augmentation method to make the classifier focus more on internal objects other than on the background. Our method first disentangles the foreground and background features. Then, we randomly swap the disentangled features within mini-batches via a two-way process. These features contain various contexts that do not appear in the biased dataset, but the class relevant representation is preserved. In addition, we introduce training schemes to obtain further performance gains. Experimental results showed that when our augmentation method was used in various weakly supervised semantic segmentation methods trained on the Pascal VOC 2012 dataset, the performance of the localization maps and pseudo-mask as well as the segmentation results improved.

## 1 INTRODUCTION

Semantic segmentation is a task that classifies objects in an image in pixels. A large number of pixel-level labels are required to train a semantic segmentation network. However, acquiring such labels are costly and time consuming. To alleviate this problem, research on weakly supervised semantic segmentation (WSSS) is being actively conducted. WSSS uses a weak label that contains less information about the location of an object than a pixel-level label but has a cheaper annotation cost. Examples of such weaker forms of labels are image-level class labels(Lee et al. (2021a;b; 2022b)), bounding boxes(Khoreva et al. (2017); Lee et al. (2021c); Song et al. (2019)), points(Bearman et al. (2016); Kim et al. (2022)), and scribbles(Tang et al. (2018); Lin et al. (2016)). Among these weak labels, herein, we focus on the image-level class label that is the most accessible and has the lowest annotation cost.

Most of the research on WSSS, utilizing image-level class labels, generate pseudo-masks based on localization maps using class activation map (CAM). Therefore, the performance of the segmentation network that uses the pseudo-mask as training data is dependent on the quality of the CAM. However, the classifier trained with a class label confuses the target object with the background, which in turn generates a blurry CAM(Lin et al., 2016). This is due to the biased dataset composed of images in which the target object frequently co-occurs with a certain background context(Geirhos et al., 2020). For instance, an object corresponding to the "sheep" class always appears in "grass landscape" and the visual layout is similar in many images. Among the PASCAL VOC 2012 training datasets, more than 20% of the images with "sheep" class contain "grass landscape" as the context. The same goes for cow-grass landscape, boat-water, and aeroplane-sky pairs(Lee et al., 2022a). Therefore, a classifier trained with a biased dataset depends not only on the target object but also on the bias, such as the background contexts. Because of such misleading correlations, the classifier often assigns higher scores for background regions that are adjacent to objects or fails to activate the target object region, where such objects appear, outside of typical scenes. To mitigate this short-

coming, a data augmentation method is required to prevent the classifier from being overfitted to misleading correlations.

In this paper, we propose DisEntangled FeaTure swapping augmentation(hereafter we refer to DEFT) method, to alleviate the problem of classifier biased with misleading correlations between the target object and the background. First, we disentangle the feature representation between the target object and the background as these features are highly entangled with each other, causing the classifier to become confused between these cues. To this end, we aggregate information about the target object and background, and use this information with explicitly defined labels to train the classifiers. Then, based on the disentangled representation, in each mini-batch, we randomly swap the background representation, while the foreground representation is fixed and vice versa. The swapped representation is augmented with diverse contextual information. The classifier can focus more on internal objects because the dependency between the object and a specific background is broken. The classifier trained using this augmented representation can effectively suppress the scores on background region and in turn yield a high-quality CAM.

The main contribution of this work can be summarized as follows:

- We proposed DEFT as a method to alleviate the problem that the classifier trained with the classical augmentation method suffers from false correlations between target objects and backgrounds.

- Our proposed DEFT method operates in the feature space, does not require any heuristic decisions or re-training of the network, and can be easily added to other WSSS methods.

- When DEFT was applied to other WSSS methods, the quality of the localization map and pseudo-mask generated through the classifier increased, and the performance of the segmentation network on Pascal VOC 2012 also improved.

## 2 RELATED WORK

**Weakly Supervised Semantic Segmentation** WSSS methods that use image-level class labels generate a localization map based on the initial seed CAM, and then produces a pseudo-mask through an additional refinement process. Because the initial seed identifies only the discriminative regions in the image, numerous studies have been conducted to expand such regions. AdvCAM(Lee et al., 2021b) identified more regions of objects by manipulating the attribute map through adversarial climbing of the class scores. DRS(Kim et al., 2021a) suppresses the most discriminative region, thereby enabling the classifier to capture even the non-discriminative regions. SEAM(Wang et al., 2020) regularizes the classifier so that the differently transformed localization maps are equivalent. AMN(Lee et al., 2022b) leverages a less discriminative part through per-pixel classification.

Further, several studies have been conducted to develop feasible methods to prevent the classifier from learning misleading correlations between the target object and the background. SIPE(Chen et al., 2022) captured the object more accurately through a prototype modeling for the background. ICD(Fan et al., 2020a) includes an intra-class discriminator that discriminates the foreground and background within the same class. W-OoD(Lee et al., 2022a) utilizes out-of-distribution data as extra supervision to train the classifier to suppress spurious cues. In addition, various studies have employed a saliency map as an additional supervision or post-processing (Lee et al. (2021e); Fan et al. (2020b); Lee et al. (2019); Wei et al. (2017; 2018); Yao & Gong (2020)). Our proposed method disentangles the background information in the feature space, and thus, no additional supervision is required.

**Data Augmentation** Data augmentation aims to improve the generalization ability of a classifier for unseen data by improving the diversity of the training data. The image erasing method removes one or more sub-regions in an image and replaces them with zero or random values. Cutout(DeVries & Taylor, 2017) randomly masks a specific part of the image, and Hide-and-Seek(Singh et al., 2018) allows the classifier to seek the class relevant features after randomly hiding the patch in the image. In contrast, the image mix-based method mixes two or more images. Mixup(Zhang et al., 2017) interpolates two images and labels, and CutMix(Yun et al., 2019) replaces a certain region of an image with a patch of another image. However, because the method that uses this regional patch randomly occludes the sub-regions, including both the object and background areas, the classifier

trained with this method cannot distinguish the foreground from the context. Additionally, there is an augmentation using rich saliency information for combining patches(Kim et al. (2021b; 2020); Dabouei et al. (2021)). However, since these methods use only the salient regions, it is difficult for the classifier to learn representation for non-discriminative regions or background cues. In addition, there are methods to augment the representation in the feature space(Lim et al. (2021); Verma et al. (2019)).

In context decoupling augmentation(CDA)(Su et al., 2021), the copy-and-paste augmentation method is introduced to a WSSS task. CDA decouples the object and context by pasting the pseudo-mask obtained in the first stage to another image. However, this method uses a single class image to obtain an accurate object instance and restricts the scale of the mask. Consequently, the diversity of the augmented representation is limited. Our proposed method synthesizes features irrespective of the number or size of the object mask, and thus, it can provide a more diverse representation to the classifier.

# 3 METHOD

## 3.1 MOTIVATION

Although the WSSS method generates localization maps using class labels, it is inappropriate to use augmentation in the classification task. Even if augmentation is used, the classifier might not distinguish the foreground and background cues owing to the biased dataset. We designed a toy experiment to analyze this phenomenon. First, we collected images, including the "aeroplane" class and "sky" images, which are the most frequent object–background pairs in the Pascal VOC 2012(Everingham et al., 2010) dataset. The Pascal CONTEXT(Mottaghi et al., 2014) dataset was used to determine whether "sky" was included. A total of 437 images of the aeroplane–sky pair were used for the toy experiment.

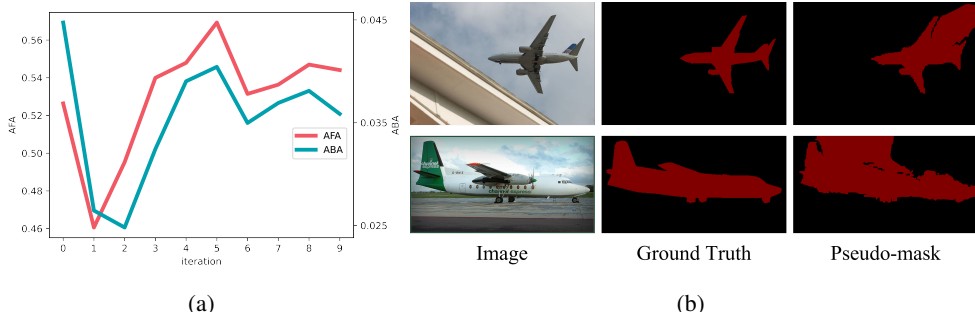

Figure 1: (a) Plot of AFA and ABA values for each iterations (b) Examples of generated pseudo-mask when using CutMix

ResNet-50(He et al., 2015) pretrained with imagenet(Deng et al., 2009) was trained on the Pascal VOC 2012 dataset after applying the CutMix augmentation(Yun et al., 2019), which is mainly used in image classification tasks. Then, we extract the feature map before the last batchnorm layer in the network for each iteration. This is to prevent the activation value of the feature map from being altered due to the affine parameters of batchnorm. Next, after dividing all pixels in the feature map into foreground and background regions using the pixel-level labels provided by the Pascal VOC 2012 dataset, we calculate the average activation value of each region, which are the average foreground activation (AFA) and average background activation (ABA), respectively. Since images contain objects of various sizes in our experiments, the AFA and ABA values are invariant to the object's scale. By analyzing trends in the AFA and ABA values, it is possible to determine whether foreground and background regions are spuriously correlated. As the number of training iterations increases, the classifier learns to discriminate between the foreground and background regions; that is, in the case of a well-trained classifier, when AFA increases, ABA decreases. However, in the case of the classifier biased by the spurious cues, the ABA does not decrease even when AFA increases. Figure 1 (a) shows the AFA and ABA values for each iteration for the toy experiment. Evidently, when the AFA value increases, the ABA value shows a similar trend for increasing and decreasing.

This trend implies that the values of the target object and background regions are highly correlated. Figure 1 (b) is an example of a pseudo-mask generated by a classifier trained with CutMix, and it can be seen that the sky, which is the background, is erroneously identified as an aeroplane. We can confirm that the object and background features are spuriously correlated. The toy experiment shows that when a pseudo-mask is generated using augmentation for the classification task, the classifier is biased with misleading correlations.

## 3.2 DISENTANGLED FEATURE SWAPPING AUGMENTATION

**Disentangling foreground and background representation.** In Section 3.1, we confirmed that classical augmentation is not suitable for generating high-level localization maps. Further, the CDA(Su et al., 2021) available in the WSSS task entails a complicated procedure for selecting the optimal mask to paste. Therefore, we propose DEFT, which is heuristic-free augmentation method suitable for generating localization maps. Our proposed method disentangles the foreground and background-related features and then swaps them between different training samples. First, we input the image $x$ into the backbone network to obtain a feature vector $z$. In general, global average pooling (GAP) is used when aggregating features. However, if GAP is used, then a coarse localization map can be generated by summarizing even the bias attributes. For this reason, several existing studies have proposed a new pooling method(Lee et al. (2021a); Araslanov & Roth (2020); Zhu et al. (2021)). Inspired by previous research, we aggregate foreground and background by different aggregators $M_{fg}, M_{bg}$ respectively. We compute the foreground and background-related attention maps from the output of the backbone network, and then use them to aggregate information respectively. First, for the attention map calculation, we utilize output $z \in \mathbb{R}^{N \times C \times HW}$ from the backbone network. In this case, $N, C, HW$ each denotes the batch size, the output channel and the spatial dimension. For output $z$, we implement different 1x1 convolution functions $\theta(\cdot)$ and $\phi(\cdot)$ to reduce the input dimension. $W_\theta \in \mathbb{R}^{1 \times 1 \times M \times C}, W_\phi \in \mathbb{R}^{1 \times 1 \times M \times C}$ is the learnable kernel of each function, which encodes information about foreground and background, respectively. Afterwards, the softmax function $\sigma(\cdot)$ is calculated on the spatial dimension to capture the region where the output of each function attended for the foreground and background.

$$A_{fg} = \sigma(\theta(z; W_\theta)), A_{bg} = \sigma(\phi(z; W_\phi)) \tag{1}$$

$A_{fg} \in \mathbb{R}^{N \times M \times HW}, A_{bg} \in \mathbb{R}^{N \times M \times HW}$ are attention maps that activate spatial importance for and foreground and background. To aggregate different features for each pixel in each attention map, $M$ channels are introduced. Then based on attention maps, we aggregate features corresponding to foreground and background. The final output of the aggregator computes $z$ and the attention map as follows:

$$z_{fg} = m(A_{fg} \otimes z), z_{bg} = m(A_{bg} \otimes z) \tag{2}$$

where $\otimes$ is matrix multiplication, and $m(\cdot)$ is a function that averages the features over the $M$ dimension. $z_{fg} \in \mathbb{R}^{N \times C \times 1}, z_{bg} \in \mathbb{R}^{N \times C \times 1}$ is aggregated feature for the foreground and background respectively.

Existing studies added a linear layer or used another backbone to effectively separate and learn different attributes (Zhu et al. (2021); Lee et al. (2021d)). In aligning with previous studies, we train the network by adding a classifier $f_{bg}$ that models the background in addition to the liner layer $f_{fg}$ modeling the object. We obtain four classification scores by feeding two disentangled features, $z_{fg}$ and $z_{bg}$, into different classifiers $f_{fg}, f_{bg}$. We effectively separate the two features by supervising them with different labels for each classification score. In the case of the classification score obtained by feeding $f_{fg}(z_{fg})$ to $f_{fg}$, the ground truth $y$ is used as the label. $f_{bg}(z_{fg})$ is the score obtained by inputting $z_{fg}$ to the classifier $f_{bg}$. Therefore, the predicted score acts as a negative sample for the target label, but as a positive sample for a class other than the target label, such that the inverse of $y$, i.e., $1 - y$, is the label. Considering the background as the inverse of the target object might be a naive approach, so it may not be appropriate to use $1 - y$ as a label to guide $z_{bg}$. However, the classifier $f_{bg}$ causes $z_{bg}$ to be far away from the foreground representation containing the target and non-target. Since $z_{bg}$ becomes a negative sample for all the classes when the background information is aggregated, the score $f_{fg}(z_{bg})$ is supervised with a zero-vector label

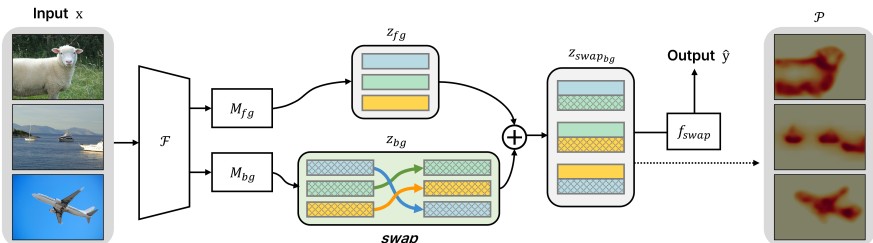

Figure 2: This is an overview of DEFT when only background feature swapping is implemented. The linear classifiers $f_{fg}$ and $f_{bg}$ learn class relevant features and class irrelevant features, respectively. The disentangled representations $z_{fg}$ and $z_{bg}$ obtained through this procedure are swapped in the mini-batch. $\hat{y}$ denotes prediction by $f_{swap}$, $\oplus$ denotes channel-wise concatenation operation and $\mathcal{P}$ denotes localization maps.

$y_0 = (0, ..., 0)$. For the classification score $f_{bg}(z_{bg})$, $z_{bg}$ is a positive sample for the background, and thus, a vector consisting of only 1 is used as the label.

$$
\begin{aligned}
\mathcal{L}_{disen} =& BCE(f_{fg}(z_{fg}), y) + \lambda BCE(f_{fg}(z_{bg}), 1 - y) \\
&+ \lambda BCE(f_{bg}(z_{fg}), 0) + \lambda BCE(f_{bg}(z_{bg}), 1)
\end{aligned}
\tag{3}
$$

During training, binary cross entropy (BCE) is used, and the loss $\mathcal{L}_{disen}$ used for feature disentangle is the same as Eq. 3. $\lambda$ is a scalar to balance between to loss terms, which were set to 0.3. $L_{disen}$ guarantees disentanglement between foreground and background features, as $z_{fg}, z_{bg}$ are trained to predicted opposite labels by two classifiers $f_{fg}, f_{bg}$. As training progresses, $z_{fg}, z_{bg}$ move away from each other in representation space due to the inverse labels.

**Swapping Disentangled Representation.** Even if the foreground and background features are effectively disentangled using the structure discussed in section 3.2, the classifier is still not free from misleading correlations because it is still trained with the biased data. Therefore, we propose a method that allows the classifier to learn representations out of the general context by swapping the disentangled features between the training sets.

We disentangle the features to be exclusive to each other with $z_{fg}$ and $z_{bg}$. $z_{fg}$ is directly related to predicting the class label, whereas $z_{bg}$ is correlated with the object, but it is not necessary to predict the class label. In other words, when the image $x$ is given, even if the background feature $z_{bg}$ is changed to another feature $z_{bg^*}$, we assumed that there should be no influence on the predictions. Therefore, the optimal classifier $f^*$ should output consistent predictions without being affected by bias.

$$
f^*([z_{fg}, z_{bg}]) = f^*([z_{fg}, z_{bg^*}])
\tag{4}
$$

That is, to achieve such invariance prediction, we implement feature swapping by randomly permuting the foreground and background features on a mini-batch in a two-way manner. The first way is to combine the class relevant attributes with the bias that do not often appear, and the second way is to combine the class irrelevant bias with the target object related attribute that does not frequently co-occur. In the first method, $\bar{z_{bg}}$ is obtained by randomly permuting the disentangled background feature, and then concatenating it with the foreground feature to $z_{swap_{bg}} = [z_{fg}, \bar{z}_{bg}]$. In the second method, the foreground features are randomly permuted to obtain $\bar{z_{fg}}$, and then the background feature is concatenated to $z_{swap_{fg}} = [\bar{z}_{fg}, z_{bg}]$. The classifier is set to predict the target label $y$ by using $z_{swap_{bg}}$ and $z_{swap_{fg}}$ as the inputs to the linear layer $f_{swap}$. In the case of $z_{swap_{fg}}$, since the target object is swapped on the mini-batch, the target label $y$ is also relocated to $\bar{y}$, according to the permuted index.

Two-way swapping enables the class relevant attribute in the feature space to be combined with the class irrelevant attribute that does not appear frequently with the corresponding class. Consequently, the classifier learns the representation that does not frequently appear in the biased dataset and is thus not misled by spurious correlations. For example, when images of an aeroplane with sky and cows with grass landscape are used as the input images, the classifier can learn representations related to

the aeroplane with grass landscape and cow appearing in sky in the feature space through two-way swapping. Unlike the CDA(Su et al., 2021), our proposed method augments the representation at the feature-level, and thus, we do not manually decide which mask to paste to decouple the object from the context. In addition, our method has the advantage of learning more diverse representations, because features in the mini-batch are combined randomly at every iteration. We design the loss $\mathcal{L}_{swap}$ for the swapped features as in Eq. 5.

$$\mathcal{L}_{swap} = BCE(f_{swap}(z_{swap_{bg}}), y) + BCE(f_{swap}(z_{swap_{fg}}), \bar{y}) \qquad (5)$$

$$\mathcal{L} = \mathcal{L}_{disen} + \mathcal{L}_{swap} \qquad (6)$$

The classifier was trained with the swapped features by adding a loss term in Eq. 3 with $\mathcal{L}_{swap}$. Eq. 6 denotes total loss function used for training the classifier. The overview of DEFT are shown in Figure 2.

**Training schemes.** We use several training schemes to improve the performance of the classifier. First, our proposed augmentation method is applied after a certain iteration. We assume that the two features are disentangled, and perform swapping. However, if the swapping is performed when the features are still entangled, then the classifier will be trained with wrong signals. The classifier preferentially learns the bias-aligned sample (e.g., boat in water), in which there is a strong correlation between the bias and labels, at the beginning of the learning process, and it learns the bias-conflicting sample (e.g., boat on a railroad), in which there is a low correlation between the bias and labels, later (Nam et al., 2020). In other words, it is difficult for the classifier to distinguish the target object related features from the bias in the early stages of learning. Therefore, the augmentation is implemented after a specific iteration $t_{aug}$, where the two features are disentangled.

In addition, we do not update the concatenated background features based on the swap loss $\mathcal{L}_{swap}$. The swapped feature is supervised by the target labels $y$ and $\bar{y}$, and if the background feature is affected by this supervision, then the feature disentanglement is not done properly. Therefore, to prevent this unintended representation learning, the corresponding feature is detached, so that the swap loss does not backpropagate to the background features $z_{bg}$ and $z_{swap_{bg}}$.

### 3.3 GENERATING PSEUDO-MASK

CAM identifies the class-relevant regions captured by the classifier within the image(Zhou et al., 2016). In general, class labels are predicted through the results of global average pooling on CAM. In this study, we fully utilize disentangled features to generate CAM. $w_{fg}$ and $w_{bg}$ are the weights of the linear layers $f_{fg}$, $f_{bg}$, respectively, and $z$ is the output of the backbone network. The localization map $\mathcal{P}$ is as follows:

$$\mathcal{P} = max(w_{fg}^T z, 1 - w_{bg}^T z) \qquad (7)$$

where $w_{fg}^T z$ is the activation map for target object and $w_{bg}^T z$ is the activation map for the background of target object. The localization map is obtained by combining both activation maps. $max(\cdot)$ is the maximum value over the channels.

We resize the original image to various scales to obtain a high resolution localization map. Since the boundary is coarse in CAM, an accurate boundary is obtained by applying the refinement method(Ahn & Kwak (2018); Ahn et al. (2019)) similar to other studies(Lee et al. (2022b; 2021b;a); Ahn et al. (2019)). We apply the IRN(Ahn et al., 2019) to the localization map to produce a pseudo-mask, which are than used for training segmentation network.

## 4 EXPERIMENT

### 4.1 EXPERIMENTAL SETUP

**Dataset and evaluation metric.** We used the Pascal VOC 2012 (Everingham et al., 2010) dataset with a total of 21 classes (20 object categories and 1 background) and 10,582 image-level class label

images augmented by (Hariharan et al., 2011) for the training. For the validation and testing, 1,449 and 1,456 pixel-level labels were used. We analyzed the performance of the generated localization map, pseudo-mask, and segmentation results through mIoU (i.e., mean intersection over union) metric, which is generally used for evaluating segmentation results.

**Implementation details.** For the classifier, we adopted ResNet-50(He et al., 2015) pretrained on ImageNet(Deng et al., 2009), and the stochastic gradient descent was used as the optimizer. The learning rate was set to 0.1 at the beginning of the training, and then decreased for every iteration by using a polynomial decay function. The classifier was trained for 10 epochs, and the point at which augmentation was applied was set to 6 through extensive experiments.

## 4.2 EXPERIMENTAL RESULT

**Quantitative results of localization maps.**

Table 1: Comparison of localization map performance when DEFT is applied to WSSS method on Pascal VOC 2012 train set.

(a) Comparison of localization map performance.

| Method | Seed |
|---|---|
| PSA(Ahn & Kwak, 2018) | 48.0 |
| + DEFT (Ours) | **51.6** |
| IRN(Ahn et al., 2019) | 48.3 |
| + DEFT (Ours) | **52.3** |
| AdvCAM(Lee et al., 2021b) | 55.6 |
| + DEFT (Ours) | **57.0** |
| (Lee et al., 2022b) | 62.1 |
| + DEFT (Ours) | **64.3** |

(b) Comparison with other augmentation methods.

| Method | Seed |
|---|---|
| w/o augmentation | 48.3 |
| Mixup(Zhang et al., 2017) | 49.0 |
| Manifold mixup(Verma et al., 2019) | 48.7 |
| Cutout(DeVries & Taylor, 2017) | 48.9 |
| CutMix(Yun et al., 2019) | 49.2 |
| CDA(Su et al., 2021) | 50.8 |
| DEFT (Ours) | **52.3** |

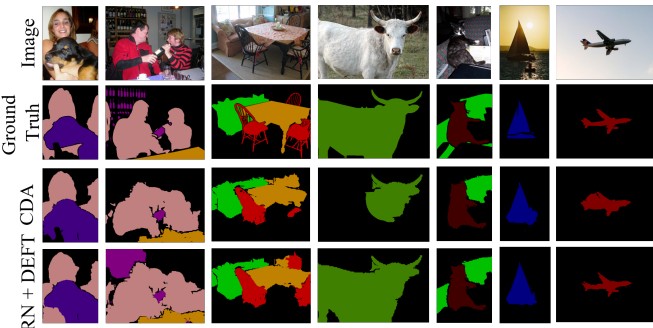

Figure 3: Examples of pseudo-masks from CDA(Su et al., 2021) and DEFT for PASCAL VOC 2012 images.

Table 1a compares the localization map obtained by WSSS baseline and DEFT. We evaluated the value mIoU(%) of the initial seed (Seed) by applying the proposed augmentation method to various WSSS methods. This localization map can be easily applied to other methods by adding the feature disentangling and swapping steps in the initial seed generation step. We applied our method to PSA(Ahn & Kwak, 2018), IRN(Ahn et al., 2019), Adv-CAM(Lee et al., 2021b), and AMN(Lee et al., 2022b). Table 1 shows the quantitative evaluation results of the localization map. The evaluation results confirmed that when the proposed method was applied, the performance improved for all the methods.

Table 1b, compares the performance of the localization maps obtained by applying different augmentations. The experimental results show that compared with the other augmentations, including the ones used in the classification and WSSS tasks, our proposed method showed the highest mIoU value. Evidently, when classical augmentation was used, the classifier becomes confused between the foreground and background cue. In addition, the result was 1.4%p higher than that obtained using the CDA(Su et al., 2021). Based on these results, it can be concluded that our proposed method enables the combination of object instances and contexts in the feature space.

**Quantitative results on pseudo-mask and segmentation network**

Table 2: Comparison of pseudo-mask and segmentation results with other WSSS methods on Pascal VOC 2012 dataset.

(a) Comparison of pseudo-mask performance on the PASCAL VOC 2012 *train* set.

| Method | Seed | Mask |
|---|---|---|
| PSA(Ahn & Kwak, 2018) | 48.0 | 61.0 |
| IRN(Ahn et al., 2019) | 48.3 | 66.3 |
| CDA(Su et al., 2021) | 50.8 | 67.7 |
| AdvCAM(Lee et al., 2021b) | 55.6 | 69.9 |
| AMN(Lee et al., 2022b) | 62.1 | 72.2 |
| PSA+ DEFT (Ours) | 51.6 | **64.2** |
| IRN+ DEFT (Ours) | 52.3 | **68.6** |
| AMN+ DEFT (Ours) | 64.3 | **72.8** |

(b) Comparison of segmentation results with other WSSS methods on Pascal VOC 2012 dataset.

| Method | val | test |
|---|---|---|
| PSA(Ahn & Kwak, 2018) | 61.7 | 63.7 |
| + DEFT (Ours) | **65.9** | **66.8** |
| IRN(Ahn et al., 2019) | 63.5 | 64.8 |
| + DEFT (Ours) | **69.1** | **68.7** |

Table 2a compares the initial seed(Seed) and the pseudo-mask(Mask) obtained by the WSSS baseline and our proposed methods. When our method was applied to different baselines(Ahn & Kwak (2018); Ahn et al. (2019); Lee et al. (2022b)), their performances showed considerable improvements in all the cases. In particular, when we applied our method on AMN(Lee et al., 2022b), an mIoU value of 72.8% was achieved for the pseudo-mask, and this value exceeded those obtained using the previous methods. In the case of IRN(Ahn et al., 2019), the result was 2.3%p ahead of the existing one. Figure 3 shows examples of the pseudo-masks generated through the CDA(Su et al., 2021), IRN, and the proposed method. These examples also show that compared to the CDA, our method captures the target object more accurately and does not mis-assign the background to the foreground.

Table 2b shows the results of training the segmentation network with the pseudo-mask obtained by applying the proposed augmentation method to PSA(Ahn & Kwak, 2018) and IRN(Ahn et al., 2019), and the evaluated mIoU value for the Pascal VOC 2012 val and test sets. The experimental results obtained using the val and test sets were 4.2%p and 3.1%p, respectively, were higher than those obtained using the PSA, and 5.6%p and 3.9%p higher than those obtained using the IRN. These observations confirmed that our proposed method can be effectively applied to the previous WSSS methods and can notably boost their performance.

## 4.3 ANALYSIS AND DISCUSSION

**Analysis for spurious correlations**

Table 3: Comparison of the mIoU values for the images with a high co-occurrence ratio in the Pascal VOC 2012 training set.

| class | aeroplane (w/ sky) | sheep (w/ grass) | cow (w/ grass) | boat (w/ water) | train (w/ railroad) |
|---|---|---|---|---|---|
| co-occurence ratio | 0.23 | 0.22 | 0.2 | 0.18 | 0.11 |
| IRN(Ahn et al., 2019) | 83.72 | 85.95 | 86.24 | 75.05 | 68.83 |
| DEFT (Ours) | **87.34** | **86.01** | **87.79** | **76.36** | **74.69** |

The mIoU values presented in Table 3 were obtained for images corresponding to frequently appearing object classes and background pairs. First, to identify the pairs that frequently appear in the training dataset, the PASCAL CONTEXT dataset was used. This dataset includes labels for various contexts, including 20 class labels of Pascal VOC 2012. We sorted the ratio of context labels that appear together by class label of Pascal VOC 2012 in descending order, and selected the most frequent pair of combinations. The target object–background combinations were aeroplane–sky, sheep–grass, cow–grass, boat–water, and train-railroad. The co-occurrence ratio denotes the ratio in which the corresponding background appears among the images corresponding to a specific class label. For example, in the entire aeroplane image, the sky coincides by more than 23%. The removal of harmful correlations was confirmed by evaluating the performance of the images of the above-mentioned combinations. All the combinations showed results than were higher than those obtained

using the IRN(Ahn et al., 2019). Thus, it can be confirmed that our proposed method can effectively alleviate the spurious correlation problem caused by a biased dataset.

**Ablations for loss terms on localization maps**

Table 4: Effectiveness of each loss on the localization map.

| Loss | (a) | (b) | (c) | (d) | Seed |
|------|-----|-----|-----|-----|------|
| baseline | ✓ | | | | 48.0 |
| $\mathcal{L}_{disen}$ | ✓ | ✓ | | | 49.8 |
| $\mathcal{L}_{swap}^{\dagger}$ | ✓ | ✓ | ✓ | | 50.9 |
| $\mathcal{L}_{swap}^{\ddagger}$ | ✓ | ✓ | ✓ | ✓ | **52.3** |

We performed an ablation study for each loss term as shown in Table 4. baseline denotes the baseline loss without the proposed augmentation method. $\mathcal{L}_{swap}^{\dagger}$ and $\mathcal{L}_{swap}^{\ddagger}$ are the first and second terms of $\mathcal{L}_{swap}$, indicating the losses for the background swapping and foreground swapping, respectively. (a) is the baseline result. The performance improvement by 1.5%p for (a) -¿ (b) implies that, benefiting from feature disentanglement, the classifier focuses more on the target object region. (b) -¿ (c) is adding $\mathcal{L}_{swap}^{\dagger}$, indicating that randomly swapping the background features is effective. The improvement along (c) -¿ (d) reflects the importance of two-way swapping.

**Manifold visualization**

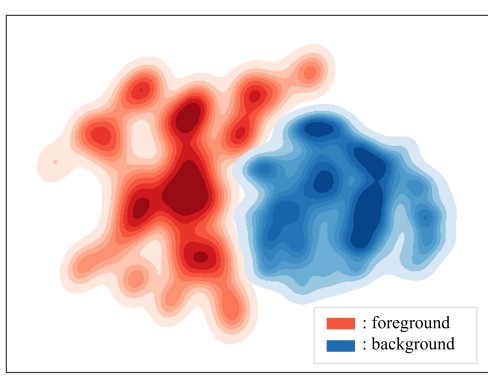

Figure 4: Visualization of the features corresponding to the target class and background by using the T-SNE(Van der Maaten & Hinton, 2008) dimension reduction method.

We performed manifold visualization by using T-SNE(Van der Maaten & Hinton, 2008), which is a dimensionality reduction method, to assess the utility of the disentangled foreground and background features. We used the intermediate features for the features $z_{fg}$ and $z_{bg}$ obtained using the aggregators $M_{fg}$ and $M_{bg}$. Figure 4 reveals that the features related to the foreground and background are semantically different.

## 5 CONCLUSION

We confirmed that the WSSS performance degrades because the classifier trained with a biased dataset that contains an image in which a specific target object frequently appears in the background utilizes background cues to identify objects. In addition, we revealed that the existing augmentation methods cannot effectively resolve this bias. To alleviate this problem, we proposed an augmentation method that disentangles the target object and background-related features in the feature space and swaps them in a mini-batch. We confirmed that the performance of the localization map and pseudo-mask, obtained through the classifier, was improved in the Pascal VOC 2012 dataset upon the application of our proposed method. Furthermore, the misleading correlation was effectively removed through the improved results obtained from the experiment performed on the class with a high co-occurrence ratio. In the future, we plan to adopt metric learning for a more effective feature disentanglement.

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
