# OpenReview forum: "Disentangled Feature Swapping Augmentation for Weakly Supervised Semantic Segmentation"
_ICLR.cc/2023/Conference — Submitted to ICLR 2023_

### Official Review · Reviewer_3SEq · 2022-10-23

**Confidence:** 4
**Correctness:** 3
**Technical Novelty And Significance:** 3
**Empirical Novelty And Significance:** 3
**Recommendation:** 5

**Clarity, Quality, Novelty And Reproducibility:**

The authors should clarify the dataset used for evaluation in Table 2 (a) in detail.

**Strength And Weaknesses:**

Strength:
1. The paper is clear and well-written.
2. The paper is well-motivated.
3. Extensive experimental results showed that when the proposed augmentation method was used in various weakly supervised semantic segmentation methods, the performance of the localization maps and pseudo-mask and the segmentation results improved.

Weaknesses:
1. The description of the dataset is not clear. For example, it is better to clarify whether the results in Table 2 (a) is evaluated on the test set or validation set.
2. The novelty is somewhat weak. There are some works[1,2,3] that use saliency-based mixup data augmentation. Please discuss the differences from these works.

[1] Kim J H, Choo W, Song H O. Puzzle mix: Exploiting saliency and local statistics for optimal mixup[C]//International Conference on Machine Learning. PMLR, 2020: 5275-5285.
[2] Kim J H, Choo W, Jeong H, et al. Co-mixup: Saliency guided joint mixup with supermodular diversity[J]. arXiv preprint arXiv:2102.03065, 2021.
[3] Dabouei A, Soleymani S, Taherkhani F, et al. Supermix: Supervising the mixing data augmentation[C]//Proceedings of the IEEE/CVF Conference on Computer Vision and Pattern Recognition. 2021: 13794-13803.


**Summary Of The Paper:**

To alleviate the problem that the classifier confuses the background with the target objects, this paper proposes to disentangle the feature representation between the target object and the background,  and swaps the background representation while the foreground representation is fixed and vice versa. The classifier trained using this augmented representation can generate more accurate localization maps.

**Summary Of The Review:**

The paper is well-motivated and clear, and the experimental results verify the effectiveness of the proposed augmentation method. However, the authors should clarify the differences from the works.
After a discussion with other reviewers, I decide to change my score to 5 since the idea of this paper is similar to Lee et al.(2021) and saliency-based argumentation methods.

---

> ### Author Response · Authors · 2022-11-18
> **Respone to Reviewer 3SEq**
>
> Thank you for your comments! We addressed your concerns as follows.
>
> **Q1) The description of the dataset is not clear. For example, it is better to clarify whether the results in Table 2 (a) is evaluated on the test set or validation set.**
>
> We apologize for the confusion. We did not clearly indicate the dataset used in the experiment. We modified the table description in Section 4.2, Table 2(a) to "Comparison of pseudo-mask performance on PASCAL VOC 2012 training set". In the experiment in Table 2(a), we used the PASCAL VOC 2012 training set as in previous studies.
>
> **Q2) The novelty is somewhat weak. There are some works[1,2,3] that use saliency-based mixup data augmentation. Please discuss the differences from these works.**
>
> [1] Kim J H, Choo W, Song H O. Puzzle mix: Exploiting saliency and local statistics for optimal mixup[C]//International Conference on
> Machine Learning. PMLR, 2020: 5275-5285.
>
> [2] Kim J H, Choo W, Jeong H, et al. Co-mixup: Saliency guided joint mixup with supermodular diversity[J]. arXiv
> preprint arXiv:2102.03065, 2021.
>
> [3] Dabouei A, Soleymani S, Taherkhani F, et al. Supermix: Supervising the mixing data augmentation[C]//Proceedings of the
> IEEE/CVF Conference on Computer Vision and Pattern Recognition. 2021: 13794-13803.
>
> The novelty of our paper is that we proposed feature-based augmentation in the weakly supervised semantic segmentation (WSSS) task, which has rarely been studied. In the case of CDA, which is the first augmentation study on the WSSS task, several heuristic decisions are required to select a pseudo-mask: how many images to use, how to size the mask to paste, etc. Additionally, the variety of images that can be combined using the cut-and-paste method is limited. Meanwhile, since our study operates at the feature level, it does not require heuristic decisions. In addition, by swapping features in a two-way manner, various representations not included in the training dataset are provided to the classifier.
> The problem with saliency-based mixup data augmentation is that it cannot use background information or non-discriminative area information because it uses a salient area. Moreover, since salient regions are combined, the classifier can only access limited representations. In contrast, our study utilizes all areas of the image, including the background, in the feature stage.

---

### Official Review · Reviewer_sGvK · 2022-10-24

**Confidence:** 5
**Clarity, Quality, Novelty And Reproducibility:** Please see above
**Correctness:** 3
**Technical Novelty And Significance:** 2
**Empirical Novelty And Significance:** 3
**Recommendation:** 3

**Strength And Weaknesses:**

Strength:

+ The paper is overall well-written and easy to understand.

+ The authors address the important problem in weakly supervised semantic segmentation.

+ The proposed technique improves several previous works.

Weakness:

- Motivation in Section 3.1 is not convincing for me.

1) The correlation between AFA and ABA seems trivial, because both values are determined by the scale of network output. The affine parameters of batchnorm determine the scale of activation scale, so AFA and ABA may be determined by those affine parameters. For example, if we split the image into two regions (left-half and right-half) and plot the average activation score of each region, the two scores will be also correlated.

2) In my opinion, the problem in Figure 1(b) is not spurious correlation, but the smoothness of CAM.

3) If the authors would like to check if the augmentation for the classification task brings misleading correlation, the baseline without any augmentation should be compared.

- The overall structure is too similar to Zhu et al. (2021). In addition, it is not clear the disentangling can be achieved as we intended. For example, if the network thinks the grass is a part of cow, z_fg can be {cow, grass} and z_bg  can be {other patterns}.

- Following recent works, experiments on the MS COCO dataset should be conducted.

- Minor points

1) If I understand correctly, in Eq (3), f_fg(z_bg) and f_bg(z_fg) should be changed.

2) In the last part of Section 3.2, the sentence "an unintended representation can be learned" is ambiguous. Can you present more details?

3) In Table 4, I recommend the authors to include improvements for all classes. In particular, I want to see the performance change for 'train' class, because 'train' has a significant correlation with 'rail'.

4) Table 3 seems strange. What is the meaning of (a, b, c, d)?

5) In Section 3.2, why M channels should be introduced? Can you present the effectiveness of values of M?

6) I want to see the results of AMN+DEFT in Table 2(a).

7) Please consider to include the T-SNE of IRN in Figure 4.

**Summary Of The Paper:**

This paper proposed a data augmentation technique for weakly supervised semantic segmentation. It is well known that previous weakly supervised semantic segmentation works have an issue of spurious correlation between foreground and background. This paper proposes the augmentation technique by swapping the disentangling features. The proposed techniques improve several previous techniques.

**Summary Of The Review:**

The paper is easy to understand and straightforward. However, my major concerns are 1) whether the disentangling can be achieved as we intended and 2) the marginal contribution over previous work.

---

> ### Author Response · Authors · 2022-11-18
> **Respone to Reviewer sGvK(2/2)**
>
> **Q6) In the last part of Section 3.2, the sentence "an unintended representation can be learned" is ambiguous. Can you present more details?**
>
> Our goal is to train $z_{bg}$ to contain only information related to the background as much as possible. However, when $z_{bg}$ is influenced by learning signals related to class labels by $L_{disen}$, it will be supervised to include foreground attributes. Therefore, unlike our previous intention, $z_{bg}$ can be entangled with foreground attribute, so we stated "an unintended representation can be learned". We revised that expression to "the feature disentanglement is not done properly".
>
> **Q7) In Table 4, I recommend the authors to include improvements for all classes. In particular,
> I want to see the performance change for 'train' class, because 'train' has a significant correlation with 'rail'.**
>
> Thank you for highlighting this point. To show that spurious correlations are effectively mitigated by DEFT, we selected object–background pairs with high co-occurring ratios and conducted experiments. We have added experimental results for the train–railroad combination in Section 4.3. As a result of the experiment, the mIoU value was improved even for the train–railroad combination with a high co-occurring ratio.
>
> |        class        | aeroplane (w/ sky) | sheep (w/ grass) | cow (w/ grass) | boat (w/ water) | train (w/ railroad) |
> |:-------------------:|:------------------:|:----------------:|:--------------:|:---------------:|:-------------------:|
> | co-occurrence ratio |        0.23        |       0.22       |       0.2      |       0.18      |         0.11        |
> |         IRN         |        83.72       |       85.95      |      86.24     |      75.05      |        68.83        |
> |      DEFT(Ours)     |      **87.34**     |     **86.01**    |    **87.79**   |    **76.36**    |      **74.69**      |
>
> **Q9) In Section 3.2, why M channels should be introduced? Can you present the effectiveness of values of M?**
>
> $M$ aggregates different features for each pixel in each attention map. By adopting an $M$ size channel, the amount of computation can be reduced by effectively reducing the dimension. Additionally, due to this dimensionality reduction, redundant features not used for deciding the objectness of the feature map can be effectively reduced.
>
> **Q10) I want to see the results of AMN+DEFT in Table 2(a).**
>
> Table 2(a) shows the results of learning the generated pseudo-mask through the segmentation network. We applied DEFT to AMN to generate a pseudo-mask. Based on the results of the experiment, the mIoU value was improved by 0.6%p, confirming that the proposed method was effective. However, experiments on the pseudo-mask are currently underway, and we will share the results of applying our proposed augmentation to AMN in the camera-ready version.
>
> **Q11) Please consider to include the T-SNE of IRN in Figure 4.**
>
> We performed manifold visualization through T-SNE using $z_{fg}$ and $z_{bg}$ extracted in the initial seed-generation step. We extracted foreground and background features $z_{fg}$ and $z_{bg}$ through aggregators. Our proposed method aims to model not only object-related attributes but also context-related attributes. However, IRN does not explicitly aggregate background-related information when generating an initial seed, so only implicit features for the background can be obtained. We felt that it was difficult to visualize this in the same way as our proposed method.

---

> ### Author Response · Authors · 2022-11-18
> **Respone to Reviewer sGvK(1/2)**
>
> We thank the reviewer for the positive feedback. We respond to each of your comments as follows.
>
> **Q1) Motivation in Section 3.1 is not convincing for me. The correlation between AFA and ABA seems trivial, because both values are determined by the scale of network output. The affine parameters of batchnorm determine the scale of activation scale, so AFA and ABA may be determined by those affine parameters. For example, if we split the image into two regions (left-half and right-half) and plot the average activation score of each region, the two scores will be also correlated. In my opinion, the problem in Figure 1(b) is not spurious correlation, but the smoothness of CAM.**
>
> Thank you for this suggestion. As the reviewer pointed out, we extracted the feature maps from the penultimate layer before the last batchnorm layer. The extracted feature maps are free from the effect of batchnorm's affine parameter. Therefore, the ABA/AFA value can be effectively used as a metric to determine whether the classifier is affected by misleading correlation. Based on the toy experiment, when using Cutmix, the trends in the AFA/ABA values reveal that the foreground and background features are highly correlated. Additionally, we replaced CAM images with pseudo-mask images generated through Cutmix to more effectively show the effect of misleading correlation in Figure 1 (b). As shown in the figure, the sky area adjacent to the aeroplane is incorrectly identified as the aeroplane.
>
> **Q2) If the authors would like to check if the augmentation for the classification task brings misleading correlation, the baseline without any augmentation should be compared.**
>
> We appreciate the reviewer's feedback. In Table 1(a), we added the mIoU value of the inital seed when augmentation was not used.
>
> |      **Method**      | **Seed** |
> |:--------------------:|:--------:|
> | **w/o augmentation** | **48.3** |
> |         Mixup        |   49.0   |
> |    Manifold mixup    |   48.7   |
> |        Cutout        |   48.9   |
> |        CutMix        |   49.2   |
> |          CDA         |   50.8   |
> |    **DEFT(Ours)**    | **52.3** |
>
> **Q3) The overall structure is too similar to Zhu et al. (2021). In addition, it is not clear the disentangling can be achieved as we intended. For example, if the network thinks the grass is a part of cow, z_fg can be {cow, grass} and z_bg can be {other patterns}.**
>
> The core of our proposed method is feature swapping. The goal of DEFT is to cause classifiers that have learned false correlations with the training dataset to focus on the target object itself, not the background. To this end, feature swapping is performed in a two-way manner to provide the classifier with various representations that could not be encountered with the training dataset alone. This makes the classifier less dependent on the background cue. Performing disentanglement, Zhu et al.’s (2021) method alone cannot achieve these goals. This can also be confirmed by Table 4 in Section 4.3. With only $L_{disen}$, the mIoU value of the initial seed is similar to the result of applying the augmentation used in existing classification tasks, such as Cutmix (Table 1(b)). However, by performing two-way swapping, the mIoU value was significantly improved. The part we want to emphasize is "swapping" rather than "disentangle".
> $L_{disen}$ additionally penalizes when disentangled features contain unintended information. For example, as the reviewer noted, if information related to grass is included in $z_{fg}$, $f_{bg}(z_{fg})$ acts as a penalty term because it has a high logit value for $f_{bg}$ when information related to the background is included. However, since the target label of $f_{bg}(z_{fg})$ is 0, the more background-related information is entangled, the more penalties are awarded. As training proceeds, z_fg implies only {cow} as much as possible, and z_bg implies {grass, other patterns}. Thus, disentanglement via $L_{disen}$ works as we intended.
>
> **Q4) Following recent works, experiments on the MS COCO dataset should be conducted.**
>
> We are currently running experiments on the COCO dataset. As the number of classes has increased to 80, we are carefully experimenting with disentangling methods and feature swapping in various ways. We will share the results of applying our proposed augmentation method to COCO dataset in the camera-ready version.
>
> **Q5) If I understand correctly, in Eq (3), f_fg(z_bg) and f_bg(z_fg) should be changed.**
>
> The equation is correct as is because, in $f_{fg}(z_{bg}), z_{bg}$ is negative for the classifier $f_{fg}$, which classifes the class label. So, we assigned an inverse of target label to it. For $f_{bg}(z_{fg}), z_{fg}$ is negative for the classifier $f_{bg}$, which classifies the background. Thus, we assign 0 as the target label.

---

### Official Review · Reviewer_vkH3 · 2022-10-25

**Confidence:** 4
**Correctness:** 2
**Technical Novelty And Significance:** 3
**Empirical Novelty And Significance:** Not applicable
**Recommendation:** 5

**Clarity, Quality, Novelty And Reproducibility:**

The paper is written clearly, so it is not difficult to implement the proposed method even if the code is not provided. The novelty exists to some extent, but performance comparison is insufficient.

**Strength And Weaknesses:**

### Strengths
1. The paper is clearly written and easy to follow with detailed equations and figures.
2. Many analyzes were performed to explain motivation and results.
\
&nbsp;

### Weaknesses

**1. Motivation** \
It is hard to understand the motivation (in Section 3.1) by Figure 1. The authors claim that backgrounds highly correlated with objects are included in pseudo groundtruth. However, it seems that the pseudo groundtruth is not just fine-grained and contains the surrounding area of the object. When we see the CAM results, the values are not high in the background that is far away from the object. For Figure 1a, I guess the reason ABA and AFA have similar values is that the surrounding pixels are included in proportion to the object's size. Also, for the qualitative results of the baseline method in Figure 3, it is difficult to find a case where the background far away from the object is recognized as a foreground class.

**2. Disentanglement loss (Eq. 3)** \
Basically, it is hard to understand why this loss can disentangle the fg and bg features well. As with existing methods, even if a highly correlated background (for a specific class) is included in the fg feature, this loss seems to work properly. Of course, I agree that the proposed architecture has the potential to operate in the ideal direction (as the authors claimed). However, the training method seems insufficient to make this possible.

Additionally, the following questions exist.
- Why is (1-y) the target for the result of f_fg(z_bg)? I think there should be no information for the foreground objects in the background feature, so it is hard to understand why f_fg should predict as 1 for classes that are not in the image.
- Also, why do y_zero or y_one have to be vectors with the same shape as y? f_bg is a classifier that simply predicts whether the input feature is background or not, so it seems the scalar value is sufficient as an output.

**3. Performance comparison** \
Performance comparison with the recent methods is lacking. The mIoU values for the actual WSSS task are summarized in Table 2b, but only two outdated methods are listed. (Also, only IRN (2019) is compared in Table 4.) The performance of the proposed method (68.7) is far behind the SOTA methods below.
- RCA + EPS: 72.8 [A]
- Puzzle- CAM: 72.2 [B]
- SPML: 71.6 [C]

[A] Regional Semantic Contrast and Aggregation for Weakly Supervised Semantic Segmentation, CVPR 2022. \
[B] Puzzle-CAM: Improved localization via matching partial and full features, ICIP 2021. \
[C] Universal Weakly Supervised Segmentation by Pixel-to-Segment Contrastive Learning, ICLR 2021.
\
&nbsp;

### Other Comments
“For the evaluation and testing, 1,449 and 1,456 pixel-level labels were used.”: evaluation &rarr; validation

**Summary Of The Paper:**

This paper proposes a feature augmentation method to improve the performance of the WSSS task. The problem with the existing methods is that the pseudo groundtruth for the specific foreground (fg) class also includes information of a highly correlated background (bg). In order to completely separate this background information, the authors divide the features and classifiers for fg and bg and make a loss for each combination. As an augmentation method, these separated fg/bg features are swapped between images in the mini-batch. The proposed method was applied to existing WSSS methods to improve performance.

**Summary Of The Review:**

There is weak evidence on the motivation and method claimed by the authors. Also, the performance comparison was not made sufficiently. As a result, my initial rating is weak reject.

---

> ### Author Response · Authors · 2022-11-18
> **Respone to Reviewer vkH3(2/2)**
>
> **Q5) Also, why do $y_{zero}$ or $y_{one}$ have to be vectors with the same shape as $y$? $f_{bg}$ is a classifier that simply predicts whether the input feature is background or not, so it seems the scalar value is sufficient as an output.**
>
> We modified the expressions $y_{zero}$ and $y_{one}$ in Section 3.2. As pointed out, to compute binary cross entropy, $y_{zero}$ and $y_{one}$ do not have to be the same shape as the label $y$. We revised the formula in Equation 3 as follows:
>
> \begin{equation}
> L_{disen} = BCE(f_{fg}(z_{fg}), y) + \lambda BCE(f_{fg}(z_{bg}), 1-y) + \lambda BCE(f_{bg}(z_{fg}), 0) + \lambda BCE(f_{bg}(z_{bg}), 1)
> \end{equation}
>
> **Q6) Performance comparison with the recent methods is lacking. The mIoU values for the actual WSSS task are summarized in Table 2b, but only two outdated methods are listed. (Also, only IRN (2019) is compared in Table 4.) The performance of the proposed method (68.7) is far behind the SOTA methods below.
> RCA + EPS: 72.8 [A]
> Puzzle- CAM: 72.2 [B]
> SPML: 71.6 [C]
> [A] Regional Semantic Contrast and Aggregation for Weakly Supervised Semantic Segmentation, CVPR 2022.
> [B] Puzzle-CAM: Improved localization via matching partial and full features, ICIP 2021.
> [C] Universal Weakly Supervised Segmentation by Pixel-to-Segment Contrastive Learning, ICLR 2021.**
>
> We appreciate your feedback. We are currently experimenting with applying our proposed DEFT to various WSSS methods. Tables 1 and 2 confirm that DEFT is effective. As shown in Table 1a, the improved results for the initial seed were confirmed for AdvCAM[1] and AMN[2], which is a SOTA method. In addition, as shown in Table 2a, the application of our proposed method to AMN results in improved pseudo-mask quality. Further, as mentioned in Section 3.1, the main contribution of our proposed method is that it mitigates the effect of misleading correlation on the classifier, as confirmed in Table 3. In the future, we will share the results of applying our proposed augmentation to various WSSS methods, including other SOTA methods.
>
> **References**
>
> [1] Lee, Jungbeom, Eunji Kim, and Sungroh Yoon. "Anti-adversarially manipulated attributions for weakly and semi-supervised semantic segmentation." Proceedings of the IEEE/CVF Conference on Computer Vision and Pattern Recognition. 2021.
>
> [2] Lee, Minhyun, Dongseob Kim, and Hyunjung Shim. "Threshold Matters in WSSS: Manipulating the Activation for the Robust and Accurate Segmentation Model Against Thresholds." Proceedings of the IEEE/CVF Conference on Computer Vision and Pattern Recognition. 2022.

---

> ### Author Response · Authors · 2022-11-18
> **Respone to Reviewer vkH3(1/2)**
>
> Thank you for the review. We appreciate your careful consideration.
>
> **Q1) It is hard to understand the motivation (in Section 3.1) by Figure 1. The authors claim that backgrounds highly correlated with objects are included in pseudo groundtruth. However, it seems that the pseudo groundtruth is not just fine-grained and contains the surrounding area of the object. When we see the CAM results, the values are not high in the background that is far away from the object. For Figure 1a, I guess the reason ABA and AFA have similar values is that the surrounding pixels are included in proportion to the object's size.**
>
> We changed the CAM image in Figure 1 (b) to a pseudo-mask image to show our motivation more clearly. When an object and background are highly correlated, high values are assigned primarily to areas adjacent to the object. The classifier assigns low values to backgrounds far from the object. This is because the corresponding region does not contain any cue related to the object, so the classifier can easily classify it as background. However, the classifier uses the area adjacent to the object as a shortcut to predict it as an object and assigns a high value to that area. Figure 1 (b) is an example of this. The sky region adjacent to the aeroplance (i.e. wings of airplane) is allocated as an object.
> In the toy experiment in Section 3.1, we selected images in which the airplane–sky combination appeared and included objects of various sizes when calculating AFA/ABA values. Therefore, the AFA/ABA values are independent of the size of the object.
>
> **Q2) Also, for the qualitative results of the baseline method in Figure 3, it is difficult to find a case where the background far away from the object is recognized as a foreground class.**
>
> As mentioned above, if an object and background are highly correlated, the area adjacent to the object is assigned a higher value. This generates a pseudo-mask, where regions close to the object are mistakenly identified as object regions. In Figure 3, in the last column, our method correctly classifies the area as sky, whereas the lower part of the airplane wing was incorrectly classified as an airplane when using CDA.
>
> **Q3) Basically, it is hard to understand why this loss can disentangle the fg and bg features well. As with existing methods, even if a highly correlated background (for a specific class) is included in the fg feature, this loss seems to work properly. Of course, I agree that the proposed architecture has the potential to operate in the ideal direction (as the authors claimed). However, the training method seems insufficient to make this possible.**
>
> $L_{disen}$ can effectively disentangle the foreground and background for two reasons. Our proposed method performs disentanglement in two steps. Therefore, if a highly correlated background attribute is included in $z_{fg}$, it will be suppressed to have a low confidence score in the attention map, and the same applies if the foreground attribute is included in $z_{bg}$. As a second step, the disentanglement loss $L_{disen}$ can be seen as using an additional penalty term that penalizes when invalid attributes (i.e. highly correlated background features for $z_{fg}$) are included. $f_{fg}$ can penalize the background feature $z_{bg}$, but since the background has no explicit label, there is a limit to effectively supervising $z_{bg}$ or assigning a penalty. To compensate for this, $f_{bg}$ is used to supervise $z_{bg}$ and at the same time penalize the foreground attribute $z_{fg}$. Due to this additional penalty, the foreground and background features will be disentangled from each other.
>
> **Q4) Why is (1-y) the target for the result of f_fg(z_bg)? I think there should be no information for the foreground objects in the background feature, so it is hard to understand why f_fg should predict as 1 for classes that are not in the image.**
>
> Assigning $(1 - y)$ to $f_{fg}(z_{bg})$ as a label can be a naive approach because it considers the inverse of the target object as the background. If only $f_{fg}$ is used, $z_{bg}$ can be located close to the representation space of the non-target object. However, these problems can be supplemented by additionally using $f_{bg}$. Since $f_{bg}$ supervises $z_{bg}$ and penalizes $z_{fg}$, the background representation will move away from the foreground space containing the target and non-target objects. As training by classifier $f_{bg}$ proceeds, the background feature will be farther away from the foreground region in the representation space, even though it will still be closer to the non-target (i.e., class corresponding to $(1-y)$) region. Therefore, as long as $f_{bg}$ is used, it is reasonable to use $(1-y)$ as a label.

---

### Official Review · Reviewer_LuUC · 2022-10-29

**Confidence:** 4
**Correctness:** 3
**Technical Novelty And Significance:** 3
**Empirical Novelty And Significance:** 3
**Recommendation:** 5

**Clarity, Quality, Novelty And Reproducibility:**

Clarity & Quality:
There are many unclear presentations:
1. "bias-aligned" is not defined. I have to resort to Lee et al. 2021d for a clearer definition.
2. I am not sure what Figure 1 (b) trying to show even with explanation in the text.
3. How’s the AFA/ABA value calculated in detail and why it shows two regions are correlated?
4. Where is f_{fg} and f_{bg} in Figure 2?

Novelty: many ideas are similar to Lee et al. (2021d), i.e., feature swapping for augmentation.

**Strength And Weaknesses:**

Strength:
+ The author provide a detailed explanation to the unresolved issue for data augmentation of current WSSS approaches (based on CAM).
+ The proposed idea is simple and reasonable. Improvements are obvious and consistent.
+ The two-way swapping idea is also interesting and reasonably improved the performance.

Weaknesses:
- The overall idea is similar to Lee et al. (2021d) despite with different tasks.
- I am not 100% convinced by the paper being motivated as "biased data leading to degraded WSSS performance". My understanding is that the CAM will attend to the background since classification also utilize the background context, so this paper proposed an approach to avoid such "leaking" and to have more accurate foreground seeds. The goal here is very different as Lee et al. (2021d) that de-bias to increase the diversity and improve the generalization. I am not sure if describing it as de-bias is a good choice here.
- There are many unclear presentation that need to be improved.

**Summary Of The Paper:**

This paper studies the phenomenal that WSSS performance will degrade with dataset biases whee a specific target object frequently appears in the background. To resolve it, the paper proposed an augmentation method that disentangles the target object and background-related features, such that the localization map will more focus on the real foreground. Experimental results show that the proposed module can largely improve the mIoUs on top of various existing approaches.

**Summary Of The Review:**

The paper proposed an augmentation method that disentangles the target object and background-related features, such that the localization map will more focus on the real foreground. The work demonstrates an interesting idea and consistent improvements on existing WSSS approaches. Still, many questions/concerns exists. I am currently in the borderline position, and would like to render a final decision upon the author's response.

Post rebuttal/Discussion:
The authors' rebuttal/revision has partially address several of my points, including the AFA/ABA descriptions, the formulations of equations. However, there are serval concerns remaining: 1. For the idea, CAM is based for classification which explores both fg and bg. Thus, I am not sure whether the CAM itself can be well-trained if they are randomized with this augmentation.  2. As pointed by other reviewers, the presentation is not perfect and still has potential to be significantly improved. 3. Performance concern, as pointed by sGvK. Thus, I'd lower my rating to 5.

---

> ### Author Response · Authors · 2022-11-18
> **Respone to Reviewer LuUC**
>
> We appreciate your consideration of our paper and your insightful reviews. We address your concerns as follows:
>
> **Q1) The overall idea is similar to Lee et al. (2021d) despite with different tasks.**
>
> There are two major differences between Lee et al. (2021d) and our proposed method. First, there is a structural difference in the implementation of disentanglement. Unlike Lee et al. (2021d), who used two networks to learn bias and target attributes, we used a single network. Foreground and background features are disentangled by two separate classifiers. Second, unlike Lee et al. (2021d), who implemented feature swapping in a single direction, we performed two-way swapping. In our study, two-way swapping is performed as background swapping, which fixes foreground features and results in permutation of background features within a mini-batch, and foreground swapping, which fixes background features and swaps foreground features within a mini-batch. By swapping features in both directions, more diverse representations are combined, and the classifier learns to focus more on the target object regardless of the background.
>
> **Q2) I am not 100% convinced by the paper being motivated as "biased data leading to degraded WSSS performance". My understanding is that the CAM will attend to the background since classification also utilize the background context, so this paper proposed an approach to avoid such "leaking" and to have more accurate foreground seeds. The goal here is very different as Lee et al. (2021d) that de-bias to increase the diversity and improve the generalization. I am not sure if describing it as de-bias is a good choice here.**
>
> We have followed your suggestion to revise the motivation of the paper. We have revised the goal of our work so that the classifier focuses more on internal objects irrelevant of background cues other than de-biasing. In particular, we revised some sentences in the Abstract, Introduction and Conclusion to make our contribution clearer.
>
> **Q3) "bias-aligned" is not defined. I have to resort to Lee et al. 2021d for a clearer definition.**
>
> \We added descriptions of "bias-aligned" and "bias-conflicting" to Section 3.3. A "bias-aligned" sample indicates data that are strongly correlated since a specific object frequently co-occurs with a specific background. Meanwhile, a "bias-conflicting" sample refers to sample in which an object appears in an unusual context, with a low degree of correlation between the background and object.
>
> **Q4) I am not sure what Figure 1 (b) trying to show even with explanation in the text.**
>
> We agree with your observation that Figure 1 (b) is confusing. We changed the figure from a CAM image to a pseudo-mask image for clarity. In Section 3.1, we show that when an augmentation method used for conventional image classification, such as CutMix, is used, the classifier often fails to distinguish the background cue from the target object. Figure 1 (b) is an example of this. A part of the sky corresponding to the background is identified as the aeroplane. Figure 1 (b) confirms that conventional augmentation is still affected by spurious correlations.
>
> **Q5) How’s the AFA/ABA value calculated in detail and why it shows two regions are correlated?**
>
> AFA and ABA indicate the degree of activation of the foreground and background regions of the feature map, respectively. We extract feature maps from the intermediate layer in the classifier at each iteration. By using the pixel-level annotation of the PASCAL VOC 2012 dataset, we can access the spatial locations corresponding to the objects and the background in the feature map. Using pixel-level annotation, we obtain the activation value of the object region and add these values. We do the same to aggregate background activation values. When $p$ is the activation value of a feature map, $F$ is the index corresponding to the object region, and $B$ is the index corresponding to the background region.The AFA/ABA value can then be obtained as follows:
>
> \begin{equation}
> AFA = {1 \over |F|} \sum_{i \in F} p_i
> \end{equation}
>
> \begin{equation}
> ABA = {1 \over |B|} \sum_{i \in B} p_i
> \end{equation}
>
> The classifier gets better at identifying target objects and distinguishing them from background cues as training progresses. Therefore, for a well-learned classifier, as learning progresses, the AFA value will increase, while the ABA value will decrease. That is, the values of AFA and ABA are inversely proportional. Conversely, the classifier affected by the spurious correlation mistakenly assigns a high activation value to the background area, and the increasing and decreasing trends of AFA and ABA are similar. Therefore, it is possible to determine the correlation between the two areas by looking at the trends of ABA and ABA.
>
> **Q6) Where is $f_{fg}$ and $f_{bg}$ in Figure 2?**
>
> We are sorry for the confusion. We only visualized the background swapping process in Figure 2 for simplicity of expression.

---

### Author Response · Authors · 2022-11-18
**Response to all reviewers**

We appreciate the reviewers' time dedicated to our work and for providing insightful and thoughtful feedback. We are also very glad that reviewers have a positive impression of our work. We revised the manuscript and conducted additional experiments taking into account the feedback suggested by the reviewers. Changes made to the manuscript are indicated in blue text. The main modifications we made are:

1. As reviewers suggested, we modified much of Section 3.1 to emphasize the motivation of our work. In particular we have added the specific process of calculating AFA/ABA and how these metric can identify spurious correlations. Furthermore, the details of toy experiment were explained. And Figure 1 (b) was modified from CAM image to pseudo-mask image to better visualize the spurious correlation between object and background features.

2. Reflecting the reviewers' opinions, we added an explanation of how disentanglement is guaranteed through the method proposed in Section 3.2. Additionally, the equation for $L_{disen}$ was modified in consideration of reviewer vkH3's feedback.

3. We performed additional experiments as requested by reviewers. As suggested by reviewer sGvK, we conducted an analysis experiment on train-railroad, an object-background combination that frequently appears in the PASCAL VOC 2012 dataset. In addition, experimental results for the baseline without augmentation were also added.

We look forward to further discussion, hoping that the revised manuscript has addressed the reviewer's concerns.

---

### Decision · Program_Chairs · 2023-01-20

**Decision:**

Reject

**Justification For Why Not Higher Score:**

The limited contribution compared to previous work, as well as methods and experiments concerns, were the main factors why acceptance is not recommended.

**Justification For Why Not Lower Score:**

N/A

**Metareview: Summary, Strengths And Weaknesses:**

This paper initially received borderline scores, and was therefore additionally discussed in a virtual meeting with reviewers. All reviewers agreed that the work addresses an important problem and overall presents a method that is easy to understand. However, multiple reviewers felt that the work does not make significant contribution compared to previous papers pointed out by the reviewers. Some concerns were also raised about the details and reasoning behind the approach, and the experimental validation. Due to these reasons, reviewers converged on a reject recommendation during the meeting and some reviewers also lowered their score. I also concur with the reviewer assessments and recommend rejection at this time.

**Summary Of Ac-Reviewer Meeting:**

Reviewers raised multiple concerns about the significance of the contribution compared to previous papers, the details and reasoning behind the approach, and the experimental validation. They found the author rebuttal to partially but not fully address each of these concerns, which I concur with. These were the main factors contributing to the reject decision, particularly the contribution with respect to previous works.